# ADVERSARIAL PRIVACY PRESERVATION IN MRI SCANS OF THE BRAIN

## ABSTRACT

De-identification of magnetic resonance imagery (MRI) is intrinsically difficult since, even with all metadata removed, a person's face can easily be rendered and matched against a database. Existing de-identification methods tackle this task by obfuscating or removing parts of the face, but they either fail to reliably hide the patient's identity or they remove so much information that they adversely affect further analyses. In this work, we describe a new class of MRI de-identification techniques that remodel privacy-sensitive facial features as opposed to removing them. To accomplish this, we propose a conditional, multi-scale, 3D GAN architecture that takes a patient's MRI scan as input and generates a 3D volume in which the brain is not modified but the face has been de-identified. Compared to the classical removal-based techniques, our deep learning framework preserves privacy more reliably without adversely affecting downstream medical analyses on the brain, including segmentation and age prediction.

## 1 INTRODUCTION

Magnetic Resonance Images (MRI) are an essential tool used both in diagnostic and research settings, but they are a privacy risk. Detailed renderings of the head can be crafted from MRI scans using techniques such as volumetric raycasting. Those renderings, when matched against facial images, can be used to infer patient identity in a type of attack already demonstrated for CT scans (Mazura et al., 2012). Commonly, MRI scans are de-identified before sharing using crude *removal-based* techniques, which seek to remove privacy-sensitive parts of the head without disturbing the brain (Figure 1). However, as we demonstrate, these techniques often fail to reliably mask the patient's identity, or they are so aggressive that they adversely affect downstream medical analyses on the brain, *e.g.* segmentation and age prediction. In this work, instead of removing potentially essential parts of the MRI scans of the head and brain, we propose to de-identify them by reshaping the privacy-sensitive regions without altering the content of medically relevant data.

Our approach is to *remodel* privacy-sensitive facial structures rather than remove them, while leaving the brain untouched. Unlike removal-based approaches, under our method the head and face exhibit realistic appearance and structure. To accomplish this, we propose a novel multi-scale volumetric Generative Adversarial Network (GAN), called C-DeID-GAN, that conditions on a convex hull of the skull extracted from the scan to be de-identified. The generator learns to synthesize MRI volumes that *preserve* medically-sensitive regions such as the brain, while non-invertibly *remodeling* privacy-sensitive characteristics such as the face from the original scan.

It is worthwhile to point out why such an approach is necessary, when methods that extract the brain – so-called *skull-stripping* methods – already exist. In short, automated measurements behave unpredictably when data is removed. As recently shown by De Sitter et al. (2020), software designed to perform measurements (*e.g.* brain segmentation or age estimation) are developed to work robustly for original data (Smith et al., 2004; Schmidt et al., 2012). If measurements are made on data *de-identified* by removal, it can result in inaccuracies or even total failure. Thus, remodeling rather than deleting the privacy-sensitive region would be desirable because it can protect privacy and at the same time ensure robustness of the downstream medical analyses.

The main contributions of this work are as follows:

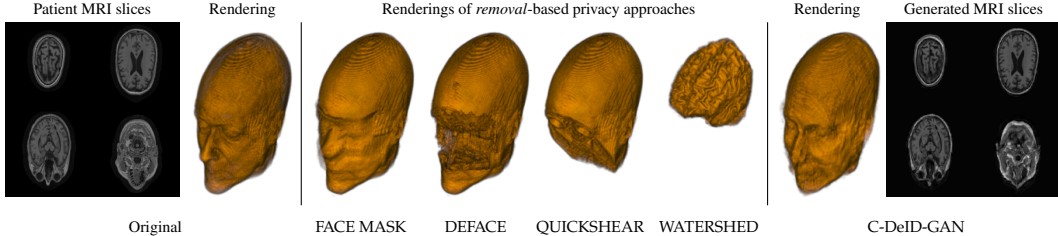

Patient MRI slices    Rendering    Renderings of *removal*-based privacy approaches    Rendering    Generated MRI slices

Original    FACE MASK    DEFACE    QUICKSHEAR    WATERSHED    C-DeID-GAN

Figure 1: *Privacy concerns in MRI scans and methods to prevent identification. (left)* Detailed 3D renderings of human heads with identifiable features can be crafted from MRI scans and used to identify patients (Mazura et al., 2012). *(center)* Existing de-identification approaches attempt to remove privacy-sensitive parts of the head, but alter the structure and appearance and often fail to reliably mask the patient's identity. *(right)* We *remodel* privacy-sensitive facial structures while leaving the brain untouched using a conditional multi-scale volumetric GAN.

1. We define a novel methodology to ensure privacy in medical imagery that enables the sharing of data in which medically relevant regions are preserved and privacy-sensitive regions are de-identified realistically

2. We propose C-DeID-GAN, a conditional multi-scale volumetric GAN that realizes a solution to the aforementioned methodology

3. We show that C-DeID-GAN preserves privacy in MRI scans more reliably than removal-based techniques without adversely affecting downstream analyses.

In addition, we make technical contributions towards the generation of the convex hull and surface representations necessary for the privacy conditioning of the GAN.

## 2 RELATED WORK

A handful of de-identification techniques exist for MRI data, which are conventionally used for sharing and distribution. The most common include *removal*-based approaches shown in Figure 1, FACE MASK (Milchenko & Marcus, 2013) DEFACE (Bischoff-Grethe et al., 2007), QUICKSHEAR (Schimke et al., 2011), and MRI WATERSHED (Ségonne et al., 2004) all of which we describe in the *Appendix*. De Sitter et al. (2020) were the first to report that these methods should be used with caution as they remove regions of the face expected by algorithms for brain segmentation and other tasks. As already pointed our in the introduction, failure modes include estimations being inaccurate or, in the worst case, it might even be impossible to perform the measurement at all. These de-identification approaches are relatively primitive and a more modern approach is currently lacking in the literature. However, Shin et al. (2018) recently proposed a *pix2pix*-inspired (Isola et al., 2016) model to generate synthetic abnormal MRI images with brain tumors. In this work, the authors argue that, in principle, their approach can be used to generate a completely artificial corpus where none of the scans can be attributed to actual patients. The downside is that brain data is also hallucinated which adversely affects medical analyses. In contrast, our approach de-identifies privacy-sensitive information of every patient, but fully preserves the medically relevant information.

More broadly, the literature on the removal of privacy-sensitive information from image data largely focuses on de-identification of photographs of faces (Jourabloo et al., 2015; Newton et al., 2005). Among these, *Deep Privacy* (Hukkelås et al., 2019) is the closest to our approach as it was the first to use GANs to de-identify faces. It conditions on an *a priori* binary segmentation, guiding the generator to inpaint privacy-sensitive regions while preserving insensitive regions.

Whereas *Deep Privacy* de-identifies conventional images of size $128 \times 128$, our goal is to generate much higher dimensional 3D MRI volumes at $128^3$ voxels – the equivalent of a $1448 \times 1448$ image. To identify privacy-sensitive face regions for conditional inpainting, *Deep Privacy* relies on a standard detector (Liu et al., 2015). Because a 3D analog does not exist, we develop an approach to extract a convex hull enclosing the head and mask of the brain for conditioning.

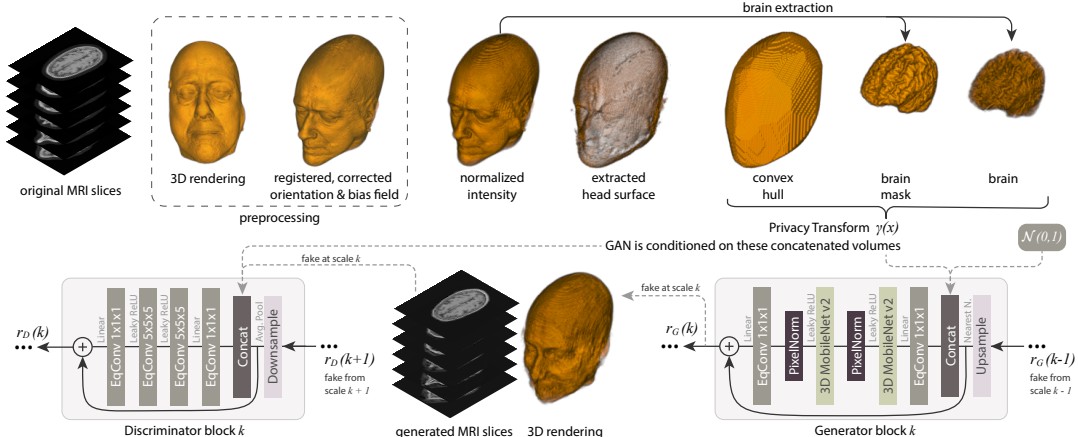

Figure 2: *Overview of our approach*: We apply model-agnostic preprocessing to standardize the scans and construct a *surface* representation using a novel technique. We then extract a convex hull $c(x)$, a brain mask $b(x)$, and the original brain intensities $b(x) \circ x$, stacked to form the privacy transform $\gamma(x)$ which serves as a conditioning variable at various scales for the generator and discriminator of our model C-DeID-GAN. It learns to convert the distribution $\mathcal{P}_X$ of original MR scans to a de-identified counterpart $\mathcal{P}_Y$, and because $\gamma(x)$ does not contain any privacy-sensitive information, renderings from the synthesized volumes are guaranteed not to reveal a patient's identity.

## 3  CONDITIONAL DE-IDENTIFICATION OF MRI VOLUMES

Given a set of MR scans $(X^{(i)})_{i=1,...N} \overset{\text{i.i.d.}}{\sim} \mathcal{P}_X$ with values in $\mathcal{I}^{S \times S \times S}$ over some intensity space $\mathcal{I} \subset \mathbb{R}$ induced by the modality[1] of the scans, we are interested in finding a function of the form $Y = f_\Phi(\gamma(X)) \sim \mathcal{P}_Y$ that maps an MR scan $X$ to its de-identified counterpart $Y$. The task of the function $\gamma(X)$ is to filter out any sensitive information in order to make it impossible to infer a patient's identity from only $Y$; to create a *privacy preserving representation*. In this work, $\gamma(X)$ is a function of the convex hull of the head $c(X) \in \{0, 1\}^{S \times S \times S}$ and the brain mask $b(X) \in \{0, 1\}^{S \times S \times S}$.

Within this *remodeling*-based privacy mapping framework, we impose three requirements:

1. *Establishment of Anonymity*: Non-invertibility of $\gamma(X)$
2. *Distribution preservation*: $\mathcal{P}_X$ and $\mathcal{P}_Y$ are stochastically indistinguishable
3. *Brain preservation*: $\forall (i, j, k) : b(X)_{i,j,k} = 1 \implies X_{i,j,k} = f_\Phi(\gamma(X))_{i,j,k}$

In other words, we are interested in deriving a function $f_\Phi$ that maps some original scan $X$ to some de-identified scan $Y$, while retaining medically relevant information (*e.g.* the brain) but preventing other information specific to $X$ to leak into $Y$ (*e.g.* the face). This makes it impossible to infer a person's identity from facial renderings alone. In the following, we describe the de-identification process depicted in Figure 2, including how to construct the privacy transform $\gamma(X)$, and how it is used to build the mapping function $f_\Phi$ via a conditional multi-scale volumetric GAN.

### 3.1  THE PRIVACY TRANSFORM $\gamma(X)$

The goal of the privacy transform is to non-invertibly change an individual MRI representation $x$ into a form $\gamma(x)$ that removes detailed privacy-sensitive information and replaces it with a convex hull filled with 1's, smoothing away detailed face information (*e.g.* eyes, nose mouth). The transform guides the GAN, showing which regions should be hallucinated via a convex hull $c(x)$ and which regions should be retained through a brain mask $b(x)$ and the brain data $b(x) \circ x$. Following common practice, we first apply a series of standard *model-agnostic* preprocessing steps to each volume including registration, denoising, and orientation correction, see the *Appendix* for details. Following the preprocessing, we define a function $c(x)$ that maps a scan $x \in \mathcal{I}^{S \times S \times S}$ to a binary convex hull

---

[1]MRI scans can be acquired under different conditions, we consider common T1-weighted MR signals.

volume of the same shape. As no efficient off-the-shelf algorithm exists, we propose a probabilistic solution that first constructs a surface representation from the MRI scan, and from this we compute the convex hull of the head. These steps are described below.

**Surface Representation.** To extract a surface representation $Z$ from an MRI scan $x$, we compute maps where rays cast from each direction intersect the head at random rotations. We then rotate these measurements back to the reference coordinates and treat each as the probability of it belonging to the surface. We begin by converting a given scan into a sequence of $K$ *binarized* and *rotated* scans

$$m^{(i)} = \mathrm{Rot}(\mathbb{1}[x \geq \delta]; R_i) \in \mathcal{I}^{S \times S \times S} \tag{1}$$

for sampled rotations $R_1, \ldots, R_K \overset{\text{i.i.d}}{\sim} \mathcal{U}(\mathrm{SO}(3))$, where $\mathcal{U}(\mathrm{SO}(3))$ denotes the uniform distribution over all rotations in three-dimensional space and $\delta \in \mathcal{I}$ represents a suitably chosen binarization threshold[2] for the binarization operator $\mathbb{1}[\cdot]$. Let us further introduce the concept of the $\zeta_{a,d}$-distance of some voxel at position $(k_0, k_1, k_2)$ for some axis $a \in \{0, 1, 2\}$ and some direction $d \in \{-1, +1\}$:

$$\zeta_{a,d}(k_0, k_1, k_2) \hat{=} \begin{cases} (S-1) - k_a & \text{if } d = +1 \\ k_a & \text{otherwise.} \end{cases} \tag{2}$$

For fixed $a$ and $d$, we can use this to create an *intersection map* $\Lambda_{a,d}^i$ for each binary image $m_S^i$:

$$\Lambda_{a,d}^{(i)}[k_0, k_1, k_2] = \mathbb{1}\left[ \left( m_{k_0,k_1,k_2}^{(i)} = 1 \right) \wedge \left( \zeta_{a,d}(k_0, k_1, k_2) = \min_{\substack{s \in \{0, \ldots, S-1\}, \\ m_{k_{a-1}|s|k_{a+1}}^{(i)} = 1}} \zeta_{q,d}(k_{a-1} \mid s \mid k_{a+1}) \right) \right] \tag{3}$$

where $(k_{a-1} \mid s \mid k_{a+1})$ indicates that the $a$-th index is set to $s$ and the two others to their associated value in $k_0, k_1, k_2$. We average the intersection map over all axis-direction combinations:

$$\Lambda^{(i)} = 1/6 \sum_a \sum_d \Lambda_{a,d}^i \tag{4}$$

This process can be thought of as casting rays from each principle direction and recording the location of the intersection with the rotated, binarized head in $m^{(i)}$. Voxels on the surface of the head will exhibit high values of $\Lambda^{(i)}$. The final step is to back-rotate $\Lambda^{(1)}, \ldots, \Lambda^{(K)}$ to the reference coordinate system and average among the $K$ randomly sampled rotations to create the *surface representation*:

$$Z = 1/K \sum_{i=1}^K Rot(\Lambda^{(i)}, R_i^{-1}) \in [0, 1]^{S \times S \times S} \tag{5}$$

Note that $Z$ is a random variable induced by the sampled rotations $R_1, \ldots, R_K$. We interpret individual voxel values of $Z$ as Bernoulli parameters characterizing the probability of some voxel belonging to the surface. This interpretation justifies binarizing $Z$ by considering it as a three-dimensional Bernoulli tensor and sampling from it on a voxel-wise basis in the next step.

**Convex Hull.** From $Z$, we sample a set of non-zero indices and use Chan's Algorithm (Chan, 1996) to compute the triangles $\mathcal{T}$ making up the convex hull. We initialize a uniform volume filled with 1's, then randomly select a sufficient number of triangles (100 suffice) from $\mathcal{T}$. For each triangle, we find its corresponding hyperplane and the half-spaces within $c(x)$ defined by it. Voxels in the outward half-space of $c(x)$ are set to 0 while the rest are unchanged, yielding a binary convex hull volume.

**Privacy Transform.** The binary convex hull volume $c(x)$ instructs the GAN as to which regions should be hallucinated. A binary brain mask $b(x)$ obtained by applying the Robex algorithm (Iglesias et al., 2011) indicates which regions should be preserved. Together, these volumes along with the masked *continuous* values of the brain $b(x) \circ x$, are concatenated to make the privacy transform $\gamma(x)$. The GAN is conditioned on $\gamma(x)$ as depicted in Figure 2 in the following subsection.

### 3.2 CONDITIONAL DE-IDENTIFICATION GAN (C-DEID-GAN)

Our proposed GAN architecture depicted in Figure 2, C-DeID-GAN, is capable of generating volumes at multiple scales and passing gradients between each scale during training. We start from a 2D

---

[2]The threshold $\delta$ is chosen to be larger than the noise values surrounding the skull.

generation framework similar to MSG-GAN (Karnewar & Wang, 2019) and adapt it to our task by means of the following: *(1)* we incorporate *conditional* information via the privacy transform, *(2)* we make architectural improvements described below, *(3)* we use a new resampling strategy, *(4)* we adopt relativistic (non-averaging) R-LSGAN loss, and *(5)* we operate on 3D volumes. We use *bottlenecks* between scales as recently suggested by Karras et al. (2019), in which the generator outputs single-channel maps instead of multi-channel maps. To reduce the memory footprint, we use modified MobileNet v2 convolutions as suggested in Howard et al. (2017).

Both the generator $G_\Phi(\gamma(x))$ and the discriminator $D_\Theta(\gamma(x), v)$ are conditioned on $\gamma(x)$, where $v$ either denotes a multi-resolutional original or fake sample. Regarding scales – suppose that $S$ and $s$ are powers of two that denote the maximum/minimum resolution synthesized by $G_\Phi$. Then both $G_\Phi$ and $D_\Theta$ are defined to have $N_B = \log_2(S/s) + 1$ blocks (indexed by $k$) that either double ($G_\Phi$) or halve ($D_\Theta$) their input resolution. Here, we generate scales from $4{\times}4{\times}4$ to $128{\times}128{\times}128$.

**Generator.** The generator $G_\Phi = G_\Phi^{(N_B)} \circ \ldots \circ G_\Phi^{(1)}$ for $G_\Phi^{(k)} : \mathbb{R}^{r_G(k-1)} \times \mathbb{R}^{r_G(k)} \to \mathbb{R}^{r_G(k)}$ and $r_G(k) = 1 \times 2^{k-1}s \times 2^{k-1}s \times 2^{k-1}s$ synthesizes a sequence of fake images $g_1, \ldots, g_{N_B}$ of increasing resolutions as follows:

$$g_k = G_\Phi^{(k)}(g_{k-1}, \gamma_k) \text{ for } k \geq 1 \tag{6}$$

where $g_0 \sim \mathcal{N}(0, 1)$ and $\gamma_k = \Downarrow_{r_G(k)} \gamma(x)$ is $\gamma(x)$ downsampled to a resolution of $r_G(k)$.

**Discriminator.** The discriminator $D_\Theta = F \circ D_\Theta^{(N_B)} \circ \ldots \circ D_\Theta^{(1)}$ for $D_\Theta^{(1)} : \mathbb{R}^{r_D(1)} \times \mathbb{R}^{r_D(1)} \to \mathbb{R}^{r_D(1)}$ resp. $D_\Theta^{(k)} : \mathbb{R}^{r_D(k-1)} \times \mathbb{R}^{r_D(k)} \times \mathbb{R}^{r_D(k)} \to \mathbb{R}^{r_D(k)}(k > 1)$ and $r_D(k) = 1 \times S/2^{k-1} \times S/2^{k-1} \times S/2^{k-1}$ assigns a *scalar* to a sequence of images[3] of decreasing resolutions $v_1, \ldots, v_{N_B}$ as follows:

$$d_k = \begin{cases} D_\Theta(v_1, \gamma_1) & k = 1 \\ D_\Theta(d_{k-1}, v_k, \gamma_k) & k > 1 \end{cases} \tag{7}$$

where $\gamma_k = \Downarrow_{r_D(k)} \gamma(x)$ is $\gamma(x)$ downsampled to a resolution of $r_D(k)$ and $F$ is a fully-connected layer that computes a scalar summary of the output of $D_\Theta^{(N_B)}$.

**Resampling blocks.** Karras et al. (2018; 2019) recently proposed to use bilinear interpolation for downsampling, but adapting this approach is problematic as it will create undesirable interpolation effects in the binary volumes. Therefore, we leverage a *probabilistic* interpretation of *average pooling* which guarantees that the proportion of non-zero voxels is preserved (in expectation) while maintaining voxel-wise correspondence to conventional average pooling performed on non-binary images. To achieve this, we first perform average pooling on the binary volume followed by interpreting the averaged values as probability parameters to voxel-dependent Bernoulli distributions. A final voxel-wise sampling step then yields the desired binary outcome (see *Appendix* for details). Upsampling is done using nearest-neighbor interpolation.

**Loss Function.** We use the relativistic (*non-averaging*) *R-LSGAN* loss (Jolicoeur-Martineau, 2018):

$$\mathcal{L}_G = 2\mathbb{E}_{(x_r, x_f) \sim (\mathcal{P}_X, \mathcal{P}_Y)} \left[ (D_\Theta(x_r) - D_\Theta(x_f) + 1)^2 \right]$$
$$\mathcal{L}_D = 2\mathbb{E}_{(x_r, x_f) \sim (\mathcal{P}_X, \mathcal{P}_Y)} \left[ (D_\Theta(x_f) - D_\Theta(x_r) + 1)^2 \right]$$

where $\mathcal{P}_X, \mathcal{P}_Y$ denote the original resp. fake distribution induced by $f_\Phi$. For simplicity, we drop the conditioning variable $\gamma(x)$ from the notation. Originally formulated for an *unconditional* scenario, we give some reasoning in the *Appendix* why the *non-averaging* class of relativistic losses is more amenable to *conditional* scenarios than the more commonly used *averaging* variants. We opt for relativistic losses as they induce a lower memory footprint than, for instance, the widely-established WGAN-GP (Gulrajani et al., 2017) requiring an additional forward/backward pass.

**Brain Preservation.** One of the requirements defined above in the Problem Definition is to perfectly preserve medically relevant information. Therefore, in a similar process to image inpainting in which original image content is masked and retained, we use the brain mask $b(x)$ to embed the original brain data into the volume synthesized by the generator: $f_\Phi(\gamma(x)) = b(x) \circ x + (1 - b(x)) \circ G_\Phi(\gamma(x))$.

---

[3] $x_1, \ldots, x_{N_B}$ in case of an original image and $g_1, \ldots, g_{N_B}$ in case of a fake image

| | | OASIS-3 | ADNI |
|---|---|---|---|
| BLURRED | | 41.28 ±1.89 | 48.03 ±1.97 |
| FACE MASK | | 34.44 ±1.79 | 42.26 ±1.88 |
| DEFACE | | 38.47 ±1.91 | 43.09 ±1.87 |
| QUICKSHEAR | | 35.25 ±1.86 | 34.28 ±1.86 |
| MRI WATERSHED | | **18.90** ±**1.54** | 22.28 ±1.56 |
| C-DeID-GAN | | 23.88 ±1.68 | **21.56** ±**1.63** |

Figure 3: *Study on De-Identification Quality: (left)* Amazon Mechanical Turk workers were asked to defeat the de-identification methods: We show the original rendering and 5 renderings of different patients, the task is to select the de-identified rendering matching the query. Here, ADNI patients de-identified using C-DeID-GAN are shown, "5" is correct. *(right)* Correct identification rates (±s.d.) for 800 distinct questions per dataset, evenly distributed among the methods. Each question was given to five workers, totalling 4,000 assignments (random guessing = 20%).

## 4 EXPERIMENTS

Above, we proposed a new and modern approach to de-identify medical image data. To judge its utility, we must address the following questions: *(1) Does remodeling preserve privacy better than existing removal-based de-identification methods? (2) Does our approach adversely affect the performance of common medical applications?* In this section, we compare our approach to other de-identification methods to answer these questions experimentally. We also show qualitatively that changing $\gamma(x)$ allows us to manipulate the synthesis of privacy-sensitive regions with C-DeID-GAN.

### 4.1 SETUP

Here, we present the datasets, the benchmark de-dentification methods that we compare our model with and the training details of our model.

**Datasets.** In this work, we use two standard publicly available large-scale Alzheimer's disease imaging studies which feature T1-weighted volumetric MR scans of the head for each subject: A selection of 2,172 MRIs from ADNI (Weiner et al., 2017; Wyman et al., 2013) and 2,168 MRIs from OASIS-3 (LaMontagne et al., 2019). Both datasets are split (80%-20% train-test) on a patient level to avoid memorizing the patient's ID from previously seen scans. Scanner types and acquisition protocols differ between and within the datasets, details can be found in the *Appendix*.

**Benchmark De-Identification Methods.** We compare our result with four publicly available and widely-established methods for de-identification of MRI head scans, depicted in Figure 1. All methods have in common that they *(1)* are not deep-learning-driven, *(2)* require no additional training and *(3)*, are used on a day-to-day basis in neuroscience and clinical research. All procedures were applied with default settings on images of resolution 128×128×128. The methods include QUICKS-HEAR (Schimke et al., 2011), FACE MASK (Milchenko & Marcus, 2013), DEFACE (Bischoff-Grethe et al., 2007),and MRI WATERSHED (Ségonne et al., 2004). Descriptions of the methods are provided in the *Appendix*.

**Training.** We train C-DeID-GAN in a distributed setup on an NVIDIA DGX-1 with V100 GPUs, where two GPUs model the generator and two control the discriminator. We use the AdamP (Heo et al., 2020) optimizer with a learning rate of $2 \cdot 10^{-3}$ and $\beta = (0, 0.99)$ and a batch size of 2. See the *Appendix* for a complete list of the hyperparameters.

### 4.2 RESULTS

In this section, we present results on (1) a user study comparing the identification rate of our model with existing de-identification methods, (2) two medical imaging end tasks on brain segmentation and age prediction and (3) our model's ability to manipulate the appearance of the generated 3D volumes. In addition, we provide a comparison of execution times of the various methods in the *Appendix* and video results of the original and de-identified scans as Supplementary Material.

**Study on De-Identification Quality.** The privacy attack described in Mazura et al. (2012) used prospectively collected data, meaning the authors had access to CT scans as well as photographs of the patient's faces. Testing de-identification quality by replicating that study for MRI scans is impossible, because photographs of ADNI and OASIS-3 patients do not exist. Therefore, we conduct

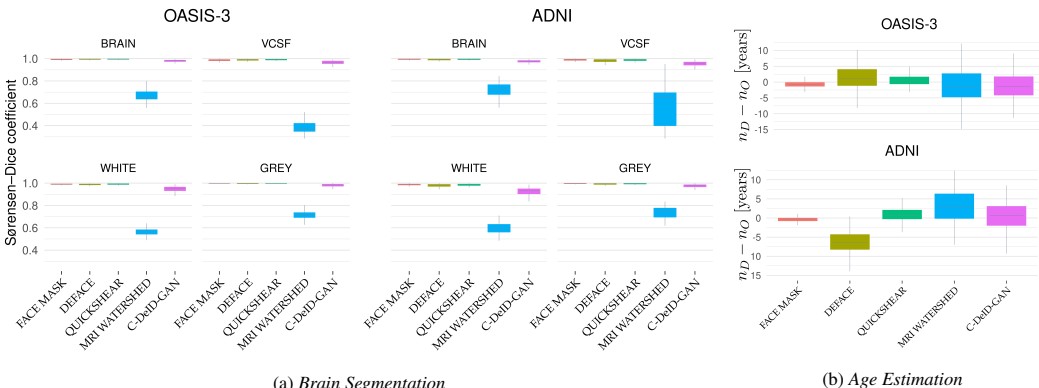

(a) *Brain Segmentation*           (b) *Age Estimation*

Figure 4: *Effect of De-Identification on Brain Segmentation and Age Estimation*: *(a)* We compare how de-identification affects the reliability of segmenting various brain regions: grey matter, white matter, VCSF, and total brain. The box plots show the Sørensen–Dice coefficient computed between segmentations on the original scan and the de-identified scan using standard software, SIENAX. Higher scores indicate better fidelity. *(b)* Box plots show how de-identification affects reliability of brain age predictions from a deep neural network. The network predicts brain age on the original scan ($n_O$) and a de-identified scan ($n_D$), and we show the distribution of $n_D - n_O$ in years. All de-identification methods show acceptable deviations, within ranges reported in other works.

a similarly-spirited study using *Amazon Mechanical Turk* in which workers are asked to defeat the various de-identification methods given renderings of MRI scans, shown in Figure 3. Workers were presented with an unaltered rendering of a query patient along with five renderings de-identified using a single method – one of which is a de-identified rendering of the query patient. The task was then to pick out the de-identified rendering which corresponds to the unaltered query rendering. The workers could choose between the five de-identified renderings of different patients, or select "uncertain". We asked 800 distinct questions per dataset. Each question was given to five workers, for a total of 4,000 assignments that are evenly distributed among the methods. The mean and the standard deviation are estimated by bootstrapping over 1,000 samples.

In the table on the right of Figure 3, we report the identification rate, or how often the workers were able to defeat each method. The identification rate accounts for the correct identifications and "uncertain" responses, see *Appendix* for details. In addition to the five de-identification methods, we added a control task BLURRED to measure the lower performance bound, in which the 2D renderings are blurred to mildly obscure the patient identity. An upper performance bound from random guessing corresponds to 20%. The results substantiate the claim that C-DeID-GAN performs extraordinarily well at de-identifying MR imagery. Although MRI WATERSHED slightly outperforms our method on OASIS-3, it removes everything but the brain, rendering it nearly useless at the downstream tasks as shown in the following sections. Our model outperforms the other de-identification methods substantially with identification rate gaps of 13%–22% on ADNI and 11%–17% on OASIS-3. We note that for both datasets, C-DeID-GAN performs near to the theoretical optimum of 20%.

**Effect of De-Identification on Brain Segmentation.** Beyond ensuring patient privacy, de-identification methods should not adversely affect software tools commonly used on MRI scans. However, it has been recently shown that certain facial de-identification methods do adversely impact subsequent automated image analysis used in research and in the clinic (De Sitter et al., 2020). In line with this study, we assess how the de facto standard brain tissue segmentation tool, SIENAX (Smith et al., 2004), performs on de-identified MRI scans in comparison to the originals. Using this tool, we segment four regions commonly used in neuroimaging: gray matter, white matter, ventricular cerebral spinal fluid (VCSF), and total brain, which can be used to assess progression in *Alzheimer's disease* (Matsuda, 2016). We record the estimated volumes and compare how segmentations obtained on the original overlaps with de-identified segmentations using the Sørensen–Dice coefficient (Sørensen, 1948; Dice, 1945), commonly referred to as *Dice score*.

In Figure 4a, we report the distribution of Dice scores between the original and de-identified scans for various brain regions. Numeric volume estimations are provided in the *Appendix*. We observe

that de-identified scans using C-DeID-GAN are reliable in comparison to the original scan, although there is a minor drop in fidelity compared to QUICKSHEAR, DEFACE and FACE MASK. In contrast, de-identification with MRI WATERSHED substantially alters the segmentation as it removes everything except the brain.

**Effect of De-Identification on Deep Learning-based Age Prediction.** Machine learning algorithms can be trained to estimate brain age from MRI scans, and the difference between predicted and chronoligcal age is shown to have links to aging and brain disease (Jónsson et al., 2019). Here, we investigate whether de-identification adversely affects brain age estimation. We train a three-dimensional adaptation of ResNet-18 (He et al., 2016) combined with a $L_2$-loss function to estimate brain age in MRI scans. Details of the training procedure can be found in the *Appendix*. We assess how the network's predicted age $n_D$ on the de-identified scans compares to the predicted age on the originals $n_O$ by measuring the difference $n_D - n_O$ between the two in years.

The results appear in Figure 4b. We find that our model performs on par with the other de-identification models, with notably little bias in the ADNI data. FACE MASK shows the least bias and smallest variance, but it is worth noting that an uncertainty of 3-4 years is typical, as chronological age is a noisy label. The deviations reported by the de-identification methods are in the range of similar age estimation studies (Huang et al., 2017; Ueda et al., 2019; Cole et al., 2017), suggesting that the effect on age estimation is acceptable. The performance of C-DeID-GAN is surprisingly good considering that the age estimation model exploits age cues in regions outside of the brain, suggesting C-DeID-GAN may implicitly model age information from the brain it conditions on.

**Controlling Synthesis via the Privacy Transform.** Finally, we demonstrate that it is possible, in principle, to control certain aspects of the synthesized MRI scan by manipulating the privacy transform $\gamma(x)$ upon which C-DeID-GAN is conditioned. As proof-of-concept, we apply a simple resizing to the privacy transformed representation, parameterized by scale factor $\alpha \in \mathbb{R}$. Recall that $\gamma(x)$ contains the brain mask, brain data, and convex hull $[b(x), x \circ b(x), c(x)]$ as depicted in Figure 5. We alter each of these volumes by resizing from shape $S^3$ to $\lfloor \alpha S \rfloor^3$, then use a linear spline interpolation to infer intensities at integral positions (where $\lfloor \cdot \rfloor$ denotes the floor function). The resulting volume is then either center cropped or evenly padded, yielding a convex hull, brain mask, and brain scaled by $\alpha$ within a volume of shape $S^3$.

In Figure 5, we observe that manipulating the size of the synthesized volume by scaling $\gamma(x)$ yields realistic results. The top three rows show the conditioning volumes from $\gamma(x)$, and the last row shows the generated MRI scans. C-DeID-GAN manages to account for the size of the conditioning information without any visible degradation of quality. We set scaling limits $(\alpha_{\min}, \alpha_{\max})$ shown in Figure 5 to reflect the distribution of brain sizes present in the data, see *Appendix* for details. As the GAN has not been exposed to $\gamma(x)$ with extreme brain sizes beyond $(\alpha_{\min}, \alpha_{\max})$ during training, scans generated at such extremes may be less realistic. This experiment demonstrates that it is possible to control the appearance of the generated volume through manipulation of the privacy transform. Future work may explore more refined means of manipulation, including some which provide even better privacy guarantees.

## 5 CONCLUSION

In this work, we have defined a new paradigm for de-identification of medical imagery and realized it for MRI scans in a novel approach, C-DeID-GAN. In contrast to traditional de-identification methods, our approach does not aim to remove certain regions but rather aims to remodel privacy-relevant information while keeping medically-relevant information untouched (*i.e.* the brain in this work). In theory, this new model class allows us produce images that appear completely genuine but do not actually contain any privacy-sensitive information. We have shown, through experiments, that our method protects privacy substantially better than existing methods without strongly affecting the performance of common tools typically found in research and clinical settings. However, we note that certain existing de-identification methods affect these downstream tasks to a lesser extent, and a future line of research will be to improve the fidelity of GAN-generated volumes to mitigate this effect. We outline these limitations and avenues for future research in the *Appendix*. The code and model weights necessary to reproduce our work will be made public upon publication. It is our hope that the approaches we have outlined in this work will find use to better preserve privacy in clinical and research settings.

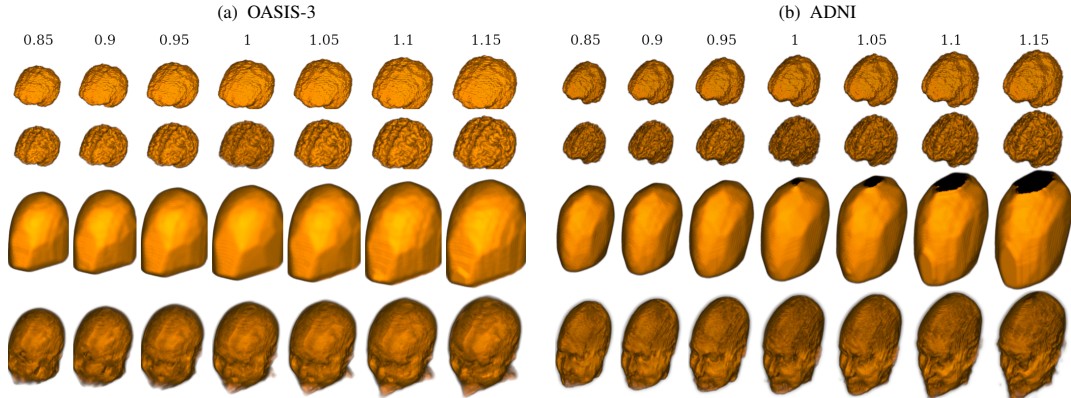

Figure 5: *Manipulating the Appearance of the Synthesis:* We demonstrate how our model can control the appearance of the synthesized volume by manipulating the privacy transform $\gamma(x)$ which C-DeID-GAN is conditioned on (top three rows). Here, we apply a simple resizing using a scale factor $\alpha \in [0.85, 1.15]$ prior to presenting it to C-DeID-GAN. Synthesized volumes (bottom row) are appropriately sized and appear realistic, implying that more sophisticated manipulations may be possible.

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
