# OpenReview forum: "Adversarial Privacy Preservation in MRI Scans of the Brain"
_ICLR.cc/2021/Conference — Reject_

### Official Review · AnonReviewer4 · 2020-10-19
**While needing to add statistical analysis, overall justified contribution in a relevant topic**

**Rating:** 7
**Confidence:** 3

**Review:**

The authors developed a GAN based method for privacy preservation in Neuroimaging called C-DeID-GAN. Providing an overview of the related work, they highlighted the shortcomings of the commonly applied methods to preserve privacy while simultaneously maintaining facial structures to enable commonly applied operations such as brain segmentation. To address these challenges the authors built on similar GANs approaches and adjusted it to the 3D dimensionality and the added criteria to fully preserve the brain area in the transformation. They experimentally validated the proposed method using two MR imaging datasets (ADNI and OASIS-3) for a) privacy preservation b) Common image processing (Brain/VCSF/GM/WM segmentation and age prediction). Finally they demonstrated a PoC for different image resolutions.

The scope of the paper is very relevant to the community. The paper is well presented within the scientific context. The methodology development and the experimental validation are well done. Overall the contribution is justified.

Major comments:

* While the experimental validation is well designed, a statistical analysis is missing and is quite essential to support some of the authors' claims. Claims such as “the model performs on par with the other models” are not meaningful enough without a statistical test to confirm it. This specifically refers to the analysis results shown in Figures 3 and 4.
* One of the main advantages of MRI compared to other neuroimaging methods is the multi-modality, i.e. utilization of different MRI sequences (such as DWI, PWI, FLAIR and others) to retrieve different properties of the brain tissue. The authors tested the operations of segmentation and age prediction but did not address this aspect. A common process that is usually done with MRI multi-modal imaging is co-registration. Where does that come into play with regard to the DeID method? How is that addressed or will be potentially addressed? I would find it useful if the authors could relate to that point.
* Regarding the analysis on controlling synthesis via the privacy transform, I’ve found the motivation for the analysis as well as the conclusion are not entirely clear. I find this analysis not directly relevant to the study objectives. I would therefore suggest moving it to the appendix and instead include the study limitations.

Minor comments:

* Page 3, Figure 2 legend line 5-6: There’s a double “to” in the sentence - please delete it.
* Page 6, Benchmark De-Identification methods: The authors state that the comparison methods are used “day-to-day basis” in the clinic. Can you support this claim by a relevant reference or other source?

---

> ### Author Response · Authors · 2020-11-20
> **Response to AnonReviewer4**
>
> We would like to thank you for the very positive feedback on our work. You have raised a couple of interesting points that we would hereby like to address.
>
> First of all we would like to clarify why we have referred to our method as being “on par” with the other methods with respect to the age estimation experiment. Age estimation is a task with underlying label noise, since the brain age doesn’t perfectly correlate with chronological age. We have found a statistical analysis to be potentially misleading to readers as the prediction errors were in the range of physiological variability for all models. We believe the performance of our method would be improved if larger datasets were available for training, or if we had access to increased computational resources.
>
> Secondly, we appreciate your comment regarding the use of multiple modalities. Initially, we intended to incorporate multiple modalities. However, we realized that modalities like T2W, FLAIR etc. are much scarcer than the T1-weighted images and the availability of consistent datasets is limited. Nonetheless, one possible approach to consider other modalities would be to use C-DeID-GAN to de-identify T1-weighted images followed by a domain-conversion network (e.g. T1W -> FLAIR) to synthesize an image in the other domains. We consider this to be a fruitful path of research that should be tackled in the future.
>
> Regarding the section on Controlling Synthesis via the Privacy Transform, the aim here was to show the principle that by manipulating $\gamma$ we can control the appearance of the generated volume. This implies there is potential for more sophisticated manipulations that might explicitly control for the shape of the cranium, the jawline, etc. In the future, we might be able to better guarantee privacy by changing appearance via manipulation of the conditioning volume. We have made adjustments to the text to explain this more clearly.
>
> Thank you for your input on the minor comments, we have addressed these in the current version of the paper. Thanks a lot for pointing them out.

---

### Official Review · AnonReviewer1 · 2020-10-21
**An interesting task but does not have wide audience in this conference.**

**Rating:** 6
**Confidence:** 4

**Review:**

The authors proposed a conditional GAN-based de-identification method for MRI scans. This method aimed to prevent possible personal information from leaking from the face surface of MRI data. The study is useful in certain clinical situations. However, the applied methods lack enough novelty in the aspect of deep learning theory. C-GAN that can manipulate the generation results have been proposed and applied to a lot of tasks in recent years. In my opinion, this submission is more suitable for medical imaging conferences such as  MICCAI, ISBI, or EMBC.

Feedbacks from authors are constructive to solve my puzzle and I decided to increase my rating.

---

> ### Author Response · Authors · 2020-11-20
> **Response to AnonReviewer1**
>
> First and foremost, we want to thank the reviewer for the helpful feedback.
>
> While we agree that the main focus of our work does not directly advance deep learning theory, this was not the objective of our work. ICLR has a broad scope, and many highly cited papers from ICLR have touched on other areas. Nevertheless, our work covers several highly relevant topics at ICLR including privacy, neuroscience, and GANs. More specifically, the conditional GAN we develop in this work contains several noteworthy innovations. Not the least of which is a new type of conditioning that must be non-invertible yet match the distribution of the original and preserve a designated region (the brain). We made other technical contributions as well, including the convex hull representation, the resampling blocks, and the use of a non-averaging relativistic loss.
>
> Finally, we note that the ICLR 2021 Call for Papers declares the following areas to be of relevance (https://iclr.cc/Conferences/2021/CallForPapers):
> >**applications in** audio, speech, robotics, **neuroscience**, computational biology, or any other field … societal considerations of representation learning including fairness, safety, **privacy**. Sister conferences ICML and NeurIPS regularly host a number of papers on medical image analysis (e.g. https://arxiv.org/pdf/2006.10511.pdf and https://arxiv.org/pdf/2006.03829.pdf at NeurIPS 2020).
>
> Putting our work in context of the recent developments in generative modeling (e.g. MSG-GAN, StyleGAN), only very recently have the techniques used in natural images become mature enough to tackle the complex problem of de-identifying MRI scans. A minimally useful volume for MRI is 128x128x128, which eclipses the largest 1024x1024 images generated by GANs today.
>
> Our paper’s main contribution is to define a new paradigm, or model class, for use in settings where part of the image must be perfectly retained but the rest must be de-identified to preserve privacy. The MRI application we present in the paper is just one example. We believe this problem will become increasingly important in the future. For example, consider uploading a face image for a skin cancer diagnosis. The face can be remodeled to preserve the patient’s privacy while the lesion is untouched.
>
> We believe that between our technical contributions and the definition of this new problem type, we have made noteworthy contributions to the ML community, showing how central problems in medical applications and society can be addressed by recent developments in deep learning. Finally, we would like to reference **AnonReviewer4** who deems our paper to be “very relevant to the community” in support of this argument.
>
> Therefore, we would like to ask you to reconsider your verdict. Let us know if you have any further remarks.

---

### Official Review · AnonReviewer2 · 2020-10-28
**ADVERSARIAL PRIVACY PRESERVATION IN MRI SCANS OF THE BRAIN**

**Rating:** 3
**Confidence:** 4

**Review:**

This paper proposes a GAN-based method to remodel the privacy information of an MRI scan, i.e., the face. They first propose a probabilistic solution to construct a surface representation and then compute the convex hull of the head. The convex hull was used to instruct the GAN to generate a synthetic face. The main contribution of this paper is that they use GAN to generate a 3D volume MRI in which the face has been remodeled rather than removed. Experiment results showed the proposed method outperforms several existing removed-based methods.
Cons:
My main concern about this paper is the purpose of remodeling the face. In my opinion, removing the original face is enough for the de-identification of an MRI scan. Why spent so much effort to generate a new one? Is the new face provides extra information to the downstream medical analyses? The authors point out that their method did not alter any content of medically relevant data. Therefore, I think remodeling the fake face is unnecessary and the method may have little practical uses.

---

> ### Author Response · Authors · 2020-11-20
> **Response to AnonReviewer2**
>
> We would like to thank you for the helpful feedback. Your concern regarding the removal of the face is indeed partially correct. It does extremely well at privacy preservation. As you can find in our experiments in the Table in Figure 3, removing everything except the brain has a 18.9% correct guess rate on OASIS-3 (beating our proposed method) and a 22.28% correct guess rate for ADNI (see results for MRI watershed, the method which removes all data except for the brain).
>
> So what is the point of retaining the face?
>
> The short answer can be found in Figure 4. Segmentation from the industry standard software fails if significant parts of the volume are missing. Age estimation also performs less reliably. **In general, automated medical measurements are developed using images with full patient data, and behave unpredictably when regions are deleted to preserve privacy.** We are not the first to realize this. For instance, [Milchenko & Marcus, 2013], the authors of FACE MASK, argue the same:
> ```
> “Many prior de-identification methods have employed pre-existing techniques that were developed for other purposes, such as MR skull stripping, a common preprocessing step in many neuroimaging studies. Skull stripping algorithms such as (Smith 2002) classify voxels into brain and non-brain, and leave only brain voxels in the dataset. This approach has the drawback that it removes anatomical features that are necessary to calculate important values such as intracranial volume and cerebrospinal fluid volume  [...]  In our experience, however, skull stripping requires careful supervision and may be affected by variation in diagnoses, age groups, MR field inhomogeneity, etc. (Fennema-Notestine et al. 2006).”
> ```
> Additionally, de Sitter et. al [2019] (referenced in “Related Work”) empirically analyzed how de-identification (denoted as FFR for “facial feature removal”) methods affect relevant clinical  and scientific downstream tasks (among those SIENAX). An excerpt of their conclusion reads as follows:
> ```
> “Our results showed that the commonly used FFR methods can lead to subsequent failures of automated volumetric pipelines. Moreover, FFR can lead to substantial changes—both random (low ICC) and systematic (significant differences)—in volumes obtained by automated methods.”
> ```
> To summarize, reshaping rather than deleting the privacy-sensitive regions is desirable because it ensures privacy and at the same time ensures robustness in the downstream automated medical analyses.
>
> Admittedly, we didn’t do a good job at outlining this thought process in the paper, as R5 was also confused on this point. Therefore, we have made changes to the **introduction and related work** in the uploaded draft to clarify this point, so that the reader can get a better understanding of the topic.

---

### Official Review · AnonReviewer3 · 2020-10-28
**This paper addresses a crucial need in medical imaging community, i.e. an automated and reliable anonymization method to remove patient’s identifiers from MRI scans. Instead of removing privacy-sensitive regions, the presented approach aims at remodeling privacy-sensitive facial structures. The experiments exhibit higher de-identification measures compare to some existing methods.**

**Rating:** 6
**Confidence:** 4

**Review:**

Pros.

    • This work develops a novel approach to de-identify privacy-sensitive regions of the MRI scans that enables sharing the data for downstream tasks like segmentation, classification etc.

    • The presented de-identification framework is non-reversible and also preserves medically sensitive regions.

    • The performance of the presented methodology is established through various simulations.


Cons.

    • While experimental results supports author’s claims regarding privacy-preservation without adversely affecting the medically relevant features, further justification is required for the superiority of the proposed method over existing algorithms. For instance, while with the DEFACE method illustrated in Figure 1 it seems impossible to identify the patient, the de-identification quality for DEFACE in Figure 3 does not agree with that. Similar argument can be made for the QUICKSHEAR method as well.

    • While it has been argued in the beginning of the manuscript that the removal-based methods can compromise the medically relevant features for the downstream tasks, this is not inferred from Figure 4 in which FACEMASK and QUICKSHEAR methods achieve higher Dice index for brain segmentation as well as lower bias (s.d.) for age estimation. Authors can improve the presentation to clarify these concerns.

---

> ### Author Response · Authors · 2020-11-20
> **Response to AnonReviewer3**
>
> First of all, we would like to thank you for your detailed feedback, your input is very insightful and helps us to improve the paper.
>
> We first address the concern about an apparent discrepancy between quantitative de-identification results in Figure 3 and qualitative results (the images depicted images in Fig. 1). When we first observed the results in Figure 3 from MTurk, we were equally surprised by the results.
>
> For this reason, we have included two pages in the Appendix [zipped in the supplementary material], one for OASIS-3 (p. 7) and one for ADNI (p. 8). **We observe that cues such as the cranial shape or shape of the ocular (eye) cavity can give hints that help the (Mechanical Turk) workers to identify patients.** These pointers do not exist, however, when it comes to C-DeID-GAN or MRI Watershed. For the former case, the privacy transform suppresses this information, and, for the latter case, the information is simply removed by the algorithm. This is exactly the reason why other methods (QUICKSHEAR, FACE MASK, DEFACE) exhibit worse results than C-DeID-GAN and MRI Watershed. Arguably, this constitutes a very strong notion of de-identification in which not only are traditional cues we would expect must be changed (eyes, nose, ears, etc.) but also less obvious ones such as cranial shape and jawline. Nonetheless, we are convinced that this more strict measure of de-identification is warranted given the significance of data-privacy within the medical field.
>
> Additionally, we would like to address the point you made about C-DeID-GAN not outperforming the other methods in every case. You are absolutely right on this matter, the rationale behind this circumstance can be explained by imperfections in the GAN quality. More specifically, we face two limitations:
> 1. Firstly, data limitations (both ADNI and OASIS-3) contain less than 3k images, which is very small in comparison to the data available for conventional GAN papers.
> 2. Secondly, hardware limitations (especially GPU memory) that make it currently impossible to make C-DeID-GAN more expressive (e.g. by adding more channels/layers)
>
> However, our main focus was on establishing a new model class (remodeling instead of removing) that helps to pave the path for more research that can further close “the gap”.
> We are grateful for you raising these points and we are devoted to adding further explanations in the paper.

---

### Official Review · AnonReviewer5 · 2020-11-04
**Confusion over point of method**

**Rating:** 3
**Confidence:** 4

**Review:**

The paper is well written, and seems to solve the proposed problem well. I am very confused however about the point of the problem you are trying to solve. From what I understand you want to take a voxelized version of an MRI scan and distort it such that the observed face is no longer identifiable with the original patient, but the useful information in the MRI is preserved. Part of your method requires or extracts the brain from the MRI. In either case you claim that that is information is available and sufficient for downstream tasks, and everything can be removed. Would a superior method here not just be to take this brain information directly.  The De-Identification Quality metric which you use would obviously get the optimal 20% accuracy here as the we have no way of associating images of brains with images of faces, and even if we do, you have not demonstrated that you method would perform well on it either. I see no point in altering the face of the scans as the face information will be useless for any downstream task, and all other information is apparently preserved, so why not just take this preserved data as the privacy preserving MRI information? While this may fail to perform well under SIENAX, I would say this is a problem with using this software as a metric, as the brain would clearly be preserved.

If I am mistaken here, I apologies, and am happy to change by rating and comments given sufficient reasoning. However, from the knowledge I posses in the field I do not think the approach has merit due to superior, trivial baselines.

I also have an issue with the experiment for assessing privacy preservation. I would say that the mechanical turk experiment is valuable, but only if you also demonstrate that deep learning based approaches cannot identify the associate the faces as well. Without this I have no way of knowing if your method fails to a simple CNN.

---

> ### Author Response · Authors · 2020-11-20
> **Response to AnonReviewer5**
>
> To begin with, we would like to show appreciation for your detailed feedback. Let us begin with the misconception about retaining only the brain. We have adjusted the text in our revision to make it more clear, since R2 was also confused on this point.
>
> To summarize the answer: reshaping rather than deleting the privacy-sensitive regions is desirable because it ensures privacy and at the same time ensures robustness in the downstream automated medical analyses. This was demonstrated in our own experiments in Figure 4 where segmentation from the industry standard software fails if significant parts of the volume are missing. Age estimation also performs less reliably. Note: MRI watershed is the method that removes all but the brain as you suggest. Beyond the results in our own study, we would like to give you a couple of examples on how other papers have justified their approach (i.e. to retain significantly more than just the brain):
> ```
> QUICKSHEAR [Schimke et al., 2011]:
> [Section 3.2]
> “Skull stripping methods are highly sensitive to parameters, which may often result into loss of desirable brain tissue. The results may also vary between methods and can require manual correction. Differences in data sets may impact further analysis, such as segmentation. Skull stripping may also favor a particular region based on the particular study [6]. This complicates meta-analysis, data re-use, and collaboration by discarding potentially relevant voxels.”
> [...]
> “It is tempting to de-identify with skull stripping since it is part of analysis, but defacing techniques allow for more flexibility. Simply skull stripping an image may discard useful data.”
> ```
> ```
> FACE MASK [Milchenko & Marcus, 2013]:
> “Many prior de-identification methods have employed pre-existing techniques that were developed for other purposes, such as MR skull stripping, a common preprocessing step in many neuroimaging studies. Skull stripping algorithms such as (Smith 2002) classify voxels into brain and non-brain, and leave only brain voxels in the dataset. This approach has the drawback that it removes anatomical features that are necessary to calculate important values such as intracranial volume and cerebrospinal fluid volume [..] In our experience, however, skull stripping requires careful supervision and may be affected by variation in diagnoses, age groups, MR field inhomogeneity, etc. (Fennema-Notestine et al. 2006).”
> ```
> In general, automated medical measurements are developed using images with full patient data, and behave unpredictably when regions are deleted to preserve privacy Most tools (c.f. SIENAX) are designed to work well on images that look like “original” MR images (i.e. not de-identified). We therefore believe that an approach that aims to retain the distribution (i.e. still looks original) has merit because it serves as a common denominator against which researchers can align their downstream models.
>
> We are a bit confused by the point that you made regarding the de-identification quality (c.f. “[...] you have not demonstrated that your method would perform well on it either”). We believe that the experiment “Study on De-Identification Quality” provides sufficient proof for claiming that C-DeID-GAN attains almost perfect de-identification results (identification rates are 23.88% [OASIS-3] resp. 21.56% [ADNI] which are both close to the theoretical optimum of 20%). We would be very open to additional input from your side to clarify this.
>
> Finally, we would like to address your point regarding a protection against algorithmic attacks (e.g. a CNN). Initially, we also envisioned to conduct an algorithmic test to check whether the synthesized images are prone to attacks. However, we quickly realized that this experiment is futile because the construction of the privacy transformation provides theoretical guarantees of privacy because only the (i) convex hull and the (ii) brain are taken from the actual patient.  The former doesn’t expose any privacy-related information while the latter needs to be present by definition. Therefore, we decided to drop this experiment in light of the strong theoretical guarantees that are induced by the proposed privacy transform.

---

### Decision · Program_Chairs · 2021-01-07
**Final Decision**

**Decision:**

Reject

**Comment:**

The paper received diverging review feedback. While reviewers found merits in the work, they also raise serious concerns over experimental validation, comparison with the existing methods, and practicality of the proposed method. It appears that the paper can benefit from better writing and more experimental validations clarifying all these points.